developmental biology, evolution, neuroscience

acoels, evolution, nervous system, regeneration, Xenacoelomorpha

**Author for correspondence:**
Mansi Srivastava
e-mail: mansi@oeb.harvard.edu

# Neural architecture and regeneration in the acoel *Hofstenia miamia*

Ryan E. Hulett[1], Deirdre Potter[2] and Mansi Srivastava[1]

[1]Department of Organismic and Evolutionary Biology, Museum of Comparative Zoology, and [2]Department of Stem Cell and Regenerative Biology, Harvard University, Cambridge, MA, USA

REH, 0000-0001-9288-7099; MS, 0000-0002-2126-8634

The origin of bilateral symmetry, a major transition in animal evolution, coincided with the evolution of organized nervous systems that show regionalization along major body axes. Studies of Xenacoelomorpha, the likely outgroup lineage to all other animals with bilateral symmetry, can inform the evolutionary history of animal nervous systems. Here, we characterized the neural anatomy of the acoel *Hofstenia miamia*. Our analysis of transcriptomic data uncovered orthologues of enzymes for all major neurotransmitter synthesis pathways. Expression patterns of these enzymes revealed the presence of a nerve net and an anterior condensation of neural cells. The anterior condensation was layered, containing several cell types with distinct molecular identities organized in spatially distinct territories. Using these anterior cell types and structures as landmarks, we obtained a detailed timeline for regeneration of the *H. miamia* nervous system, showing that the anterior condensation is restored by eight days after amputation. Our work detailing neural anatomy in *H. miamia* will enable mechanistic studies of neural cell type diversity and regeneration and provide insight into the evolution of these processes.

## 1. Introduction

Nervous systems vary in their composition and structure across animals, yet many genetic components of the nervous system are as ancient as the transition to animal multicellularity [1,2]. Despite some debate about the precise timing of their origin, animal nervous systems have a pre-bilaterian origin [3–9]. The evolution of bilateral symmetry, i.e. the origin of animals with clear anterior–posterior (AP) and dorsal–ventral (DV) axes around 550 Ma [10], corresponded to a major transition in nervous system evolution that led to the formation of (i) highly organized nervous systems that were restricted along the major body axes and (ii) an expansion of neural cell type diversity. Most bilaterian lineages, including ecdysozoans, spiralians and deuterostomes, collectively known as the Nephrozoa, have these features [11–16], and their genomes encode key enzymes for major neurotransmitter synthesis pathways (figure 1). The phylum Xenacoelomorpha (xenoturbellids, nemertodermatids, and acoels), a lineage hypothesized as the sister-group to all other bilaterians [11,12,25–27], is phylogenetically positioned to inform the evolution of metazoan nervous systems with regards to their structure and composition. In particular, functional studies of neural specification and patterning in xenacoelomorphs are needed to infer how genetic pathways underlying these processes have evolved. These studies would also inform the evolution of animal nervous systems [28–30] in the alternative scenario that places xenacoelomorphs as sister to Ambulacraria [31,32].

Studies of neural anatomy in the three major xenacoelomorph groups have shown diversity in neural structures as well as in their composition with regards to neural cell types. Xenoturbellids have an intraepidermal nerve net with no areas of condensation [17,18]. Nemertodermatids have a subepidermal nerve plexus and a ring process located at or around the statocyst, a sensory structure found in all xenacoelomorphs that is hypothesized to be involved in sensing gravity [33]. Characterized acoel nervous systems range from a low degree of

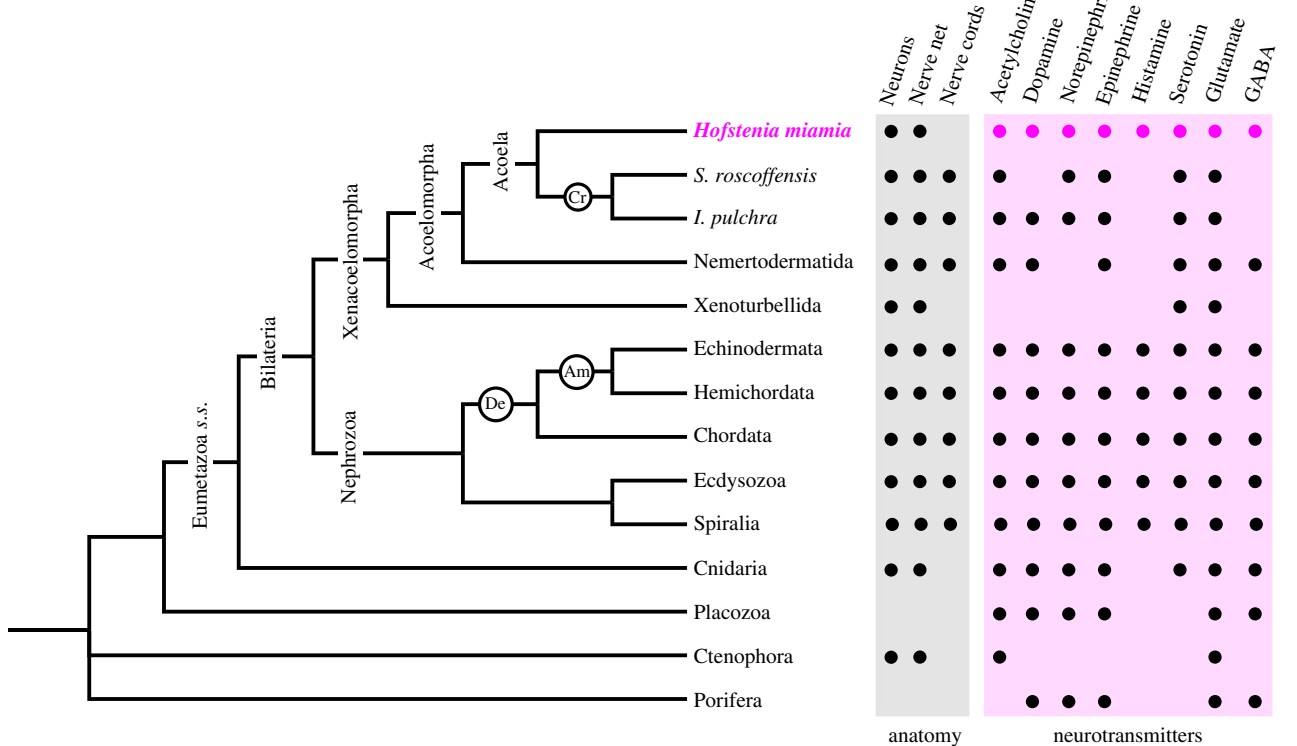

**Figure 1.** The *H. miamia* genome encodes key enzymes for all major neurotransmitter synthesis pathways. Consensus phylogenetic tree depicting the relationships of major animal groups (Eumetazoa *s.s.*: Eumetazoa *sensu stricto*; open black circles denote Cr: Crucimusculata, Am: Ambulacraria, and De: Deuterostomia) and neural features of each depicted lineage. Features coded as present if found in transcriptome or via immunohistochemistry/FISH, based on published literature (black circles) [3,17–24] or data collected in this paper (magenta circles). The absence of circles indicates feature has not been reported. (Online version in colour.)

centralization to having anterior condensations with organized structures, such as ring-shaped commissures and ventral lobes located anteriorly within the Crucimusculata, a large clade of fast-evolving acoels that includes relatively better-studied species [12,19,34–43]. Notably, all three xenacoelomorph groups show the presence of nerve nets and a consistent restriction of condensed structures along the DV axis is lacking, despite the presence of a bilaterally symmetric body plan. Genetic pathways that organize xenacoelomorph nervous systems and the functional roles of genes controlling other bilaterian nervous systems within xenacoelomorphs remain unknown.

We sought to define neural anatomy in the acoel *Hofstenia miamia*, which is amenable to mechanistic studies of nervous system development and regeneration. We analysed the *H. miamia* transcriptome [44] to identify homologues of genes from major neurotransmitter synthesis pathways. Previous work using transverse sections in *H. miamia* has shown that the nervous system is ring- or cylinder-shaped and completely encircles the pharynx in the anterior of the animal [45,46]. Our studies in *H. miamia* identified a subepidermal nerve net throughout the animal and an anterior condensation with layered organization that showed asymmetry along the DV axis. We assessed the organization of the nervous system in relationship to musculature and found that neural cells are detectable on either side of the body wall musculature, but are always internal to the peripheral longitudinal musculature. We determined the precise spatial and temporal sequence of events for regeneration of major cell types and structures in this nervous system and for rescaling of the anterior condensation relative to the body length. Our work establishes a robust framework for identification of molecular pathways that control xenacoelomorph nervous systems.

## 2. Material and methods

### (a) Animal husbandry
Animals were kept in plastic boxes at 21°C in artificial sea water, and fed brine shrimp and rotifers twice weekly, as described previously [27].

### (b) Neural gene identification and phylogenetic analysis
We queried known neurotransmitter synthesis genes from *H. sapiens*, *M. musculus*, *D. rerio* and *D. melanogaster* against the *H. miamia* transcriptome [44] to determine putative orthologues (electronic supplementary material, table S1). In some cases, phylogenetic analyses were used to establish orthology of the proteins encoded by these transcripts (electronic supplementary material, figure S1b). Protein sequences were aligned using MAFFT (v7) [47]. Alignments were trimmed using Gblocks [48] using the least stringent parameters. Phylogenetic trees were inferred using maximum likelihood analyses, with 1000 bootstrap replicates, implemented in RAxML (v8.2.4) [49] using the WAG + G model of protein evolution.

### (c) Immunohistochemistry and *in situ* hybridization
Prior to fluorescent *in situ* hybridization (FISH), animals were starved for one week and fixed in 4% paraformaldehyde in phosphate-buffered saline with 0.1% Triton (PBST) for 1 h at room temperature. Genes were amplified by PCR from cDNA (from adult animals, regenerative time points and embryonic stages)

and cloned into the pGEM T-easy vector. The FISH protocol was followed as previously described [27]. Immunohistochemistry (IHC) was performed using commercially available primary antibodies (FMRFamide; EMDMillipore AB15348, Tyrosinated tubulin; Sigma-Aldrich T9028) and custom antibodies for Tropomyosin (see section below). For detailed IHC protocol, see method in electronic supplementary material.

## (d) Custom tropomyosin antibodies

Rabbit polyclonal antibodies were custom-made by GenScript via the PolyExpress Premium using the *H. miamia* protein sequence for Tropomyosin (electronic supplementary material, table S2). Recombinant protein or protein fragments were expressed in *E. coli* using the provided protein sequence and then used to immunize two rabbits. Two antibodies (#2995 and #2997) were generated and their target binding was validated by Western Blot with the protein immunogen. Both antibodies were used together in IHC experiments at 1 µg ml$^{-1}$.

## (e) Confocal microscopy and image processing

Specimens were imaged using a Leica SP8. At least three animals were imaged per experiment. To determine co-expression, animals were imaged at 63× magnification in the anterior condensation in three different areas dorsally and two different areas ventrally. Single z-planes were examined for co-expression.

## (f) Determining the ratio of the anterior condensation length to body length

To determine the proportion of the body length occupied by the anterior condensation in intact animals and during regeneration, we performed two measurements (figure 6*b*; electronic supplementary material, table S3). The span of the anterior condensation (AC) was measured as the length of a line segment drawn from the most anterior region of the animal where *glutamate decarboxylase-1* (*gad-1*) expression was detectable to the end of the anterior condensation. The second measurement, body length (B), was the length of a line segment from the same position at the most anterior region of the animal to the tip of the tail. The AC : B ratio provided the proportion of the body length occupied by the anterior condensation. Measurements for regenerating tail fragments started 3 days post amputation (dpa), when concentrated *gad-1* expression first became observable in the anterior region.

# 3. Results

## (a) The *Hofstenia miamia* genome encodes a full complement of enzymes for major neurotransmitter synthesis pathways

We queried the transcriptome sequence data available for *H. miamia* [44] for sequences homologous to known neurotransmitter synthesis genes and assigned orthology using sequence similarity or phylogenetic analyses as appropriate (see methods, electronic supplementary material, table S1 and figure S1b). We identified putative orthologues of enzymes required for the synthesis of neurotransmitters including monoamines (serotonin, histamine, catecholamines epinephrine, norepinephrine and dopamine), acetylcholine, and the amino acid signalling molecules glutamate and GABA in the *H. miamia* transcriptome (figure 1). Many of these neural components are found in nephrozoans and in cnidarians, the outgroup lineage to bilaterians. Their presence in *H. miamia* is consistent with them being ancestral eumetazoan *sensu stricto* (cnidarian + bilaterian) characteristics. Notably, the histamine synthesis pathway was found in the *H. miamia* genome but has not been detected in non-bilaterian genomes studied thus far, suggesting it is a bilaterian innovation. The complement of neural genes shared between *H. miamia* and other bilaterians suggests that the lack of certain neurotransmitter synthesis pathways in xenoturbellids, nemertodermatids and other acoels [18,20–24,37,50] represents either a loss of genes in these lineages or a failure of the currently available genomic and transcriptomic resources in these species to detect these components.

## (b) *Hofstenia miamia* has an organized nervous system with a complex anterior condensation

To determine which structures are present within the *H. miamia* nervous system, we performed immunohistochemistry using commercially available antibodies for FMRFamide (FMRF) and tyrosinated tubulin (Tyr-tubulin), which label neurons across diverse animals. Confocal imaging from both dorsal and ventral sides identified (i) an organized, prominent neural structure located in the anterior and (ii) a subepidermal nerve net throughout the rest of the body (figure 2*a–k*). This anterior structure satisfies some definitions of a brain (e.g. it is a conglomeration of highly interconnected nerve cells [51]), but does not meet others (e.g. we did not identify an outer cortex of cell bodies with a central neuropil [50]). Therefore, in this manuscript, we refer to this brain-like structure with its many organized regions as the anterior condensation.

Immunostaining showed that the anterior condensation is composed of reticulated neurite bundles, i.e. bundles of neural processes, that are wrapped around clusters of nuclei (figure 2*b–h*). Tyr-tubulin immunostaining revealed a strongly labelled cell within each group of clustered nuclei (figure 2*g,h,i,k*). We hypothesize these cells are putatively sensory in function based on their morphology and multiciliated structure. Tyr-tubulin also labelled the frontal organ (figure 2*i,j*), a secretory and sensory structure common within acoels [52]. This structure showed multiple finger-like projections located in the anterior, near the DV margin. Another sensory structure, the statocyst, which contains a mineralized statolith surrounded by a ciliated structure and is located dorsally just above the mouth, has been reported in the previous work [27].

In a fluorescent *in situ* hybridization (FISH) screen for expression of putative neural genes, the expression of the *glutamate decarboxylase-1* (*gad-1*) gene revealed additional neural structures within the anterior condensation (figure 2*l–v*). The periphery of the anterior condensation labelled by *gad-1* marked the same reticulated neurite bundles as observed with FMRF and Tyr-tubulin immunostaining (figure 2*b–h*). This network showed no obvious asymmetries along the DV axis, connecting fully around the circumference of the anterior condensation (figure 2*m,n*). Internal to the neurite bundle network, densely packed *gad-1*$^{+}$ cells were detected (figure 2*o,p*). This population of putative neurons did not form a continuous structure along the DV span of the condensation. Although present throughout the dorsal side, *gad-1*$^{+}$ cells were absent from the ventral midline, giving the appearance of ventral lobe-like structures. The layered structure of *gad-1*$^{+}$ cells provides useful landmarks for understanding neural anatomy in *H. miamia*, and therefore, we refer to the peripheral region with neurite bundles as 'Layer I' and the internal layer of densely packed *gad-1*$^{+}$ cells as 'Layer II'.

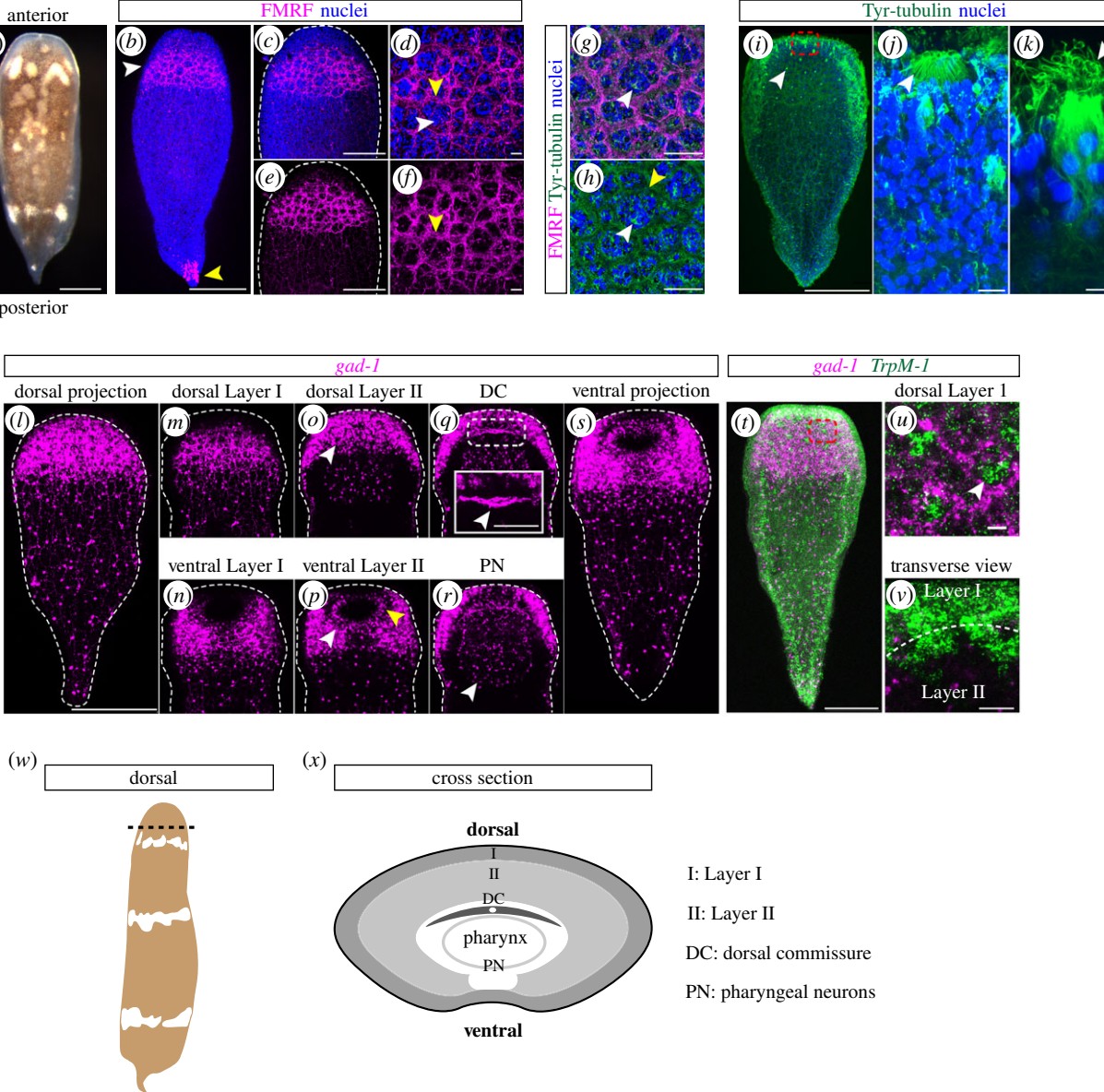

**Figure 2.** Neurons are present in high density in the anterior and are organized into discrete structures including putative sensory organs. (a) *H. miamia* juvenile animal, dorsal view. (b) FMRF immunostaining concentrated in the anterior (white arrowhead). A small number of cells in the tail tip are also labelled with this marker (yellow arrowhead). (c,d) Zoom-in of anterior showing reticulated neurite bundles with (c) and without (d) nuclei. (e,f) Further zoom-in with (e) and without (f) nuclei showing clusters of nuclei (white arrowhead) surrounded by a reticulated network of neurite bundles (yellow arrowheads). (g,h) Anti-Tyr-tubulin (green) labels neurite bundles (yellow arrowhead) labelled by anti-FMRF and cells at the centre of the nuclei clusters (white arrowheads). (i) Tyr-tubulin labels frontal organ (dashed red box; shown in higher magnification in (j)) and putative sensory cells (white arrowhead; shown in higher magnification in (k)). (j) a high magnification view of the frontal organ (white arrowhead). (k) a high magnification view of a putative sensory cell with cilia (white arrowhead). (l–s) *gad-1* mRNA expression showing layering in the anterior condensation and major neural structures. (l) Dorsal projection with *gad-1*+ cells concentrated in the anterior and diffusely distributed in the posterior. (m,n) Layer I contains a neurite bundle network. (o,p) Layer II contains mesenchyme-like cell bodies (white arrowheads). (p) Oral nerve ring located ventrally (yellow arrowhead). (q) Dorsal commissure (DC; white arrowhead) located above pharynx. Dashed white box outlines region highlighted in high magnification inset (solid white box). (r) Pharyngeal neurons (PN; white arrowhead). (s) Ventral projection with *gad-1*+ cells recapitulating the pattern seen in the dorsal view. (t) *TrpM-1* is expressed in clusters of cells throughout the body, concentrated in the anterior (dashed red box; shown in higher magnification in (u)). (u) *TrpM-1* expression in groups of clustered cells (white arrowhead) that correspond to nuclei clusters surrounded by *gad-1*+ neurite bundles within Layer I. (v) Transverse view through Layer I and Layer II (boundary between layers is denoted by dashed white line) showing that the *TrpM-1*+ cells are contiguous between the two layers. (w) A schematic of *H. miamia* showing location (dashed black line) of cross section through the anterior condensation shown in (x). (x) Schematic of cross section through the anterior condensation summarizing the neural anatomy revealed by the *gad-1* mRNA expression. Dashed white lines around specimens show the outline of the animal. Scale bars: (a,b) 200 μm, (c,d) 100 μm, (e,f) 10 μm, (g,h) 20 μm, (i) 200 μm, (j,k) 10 μm, (l–t) 200 μm, (inset in q) 60 μm, (u,v) 20 μm. (Online version in colour.)

On the dorsal side underneath Layer II, a transversely oriented *gad-1*+ neurite bundle with a ring-like shape in its centre was observed. We refer to this structure as the 'dorsal commissure' because this bundle was present dorsal to the pharynx and it extended between the left and right sides of the animal (figure 2q, inset). The pharynx, located ventrally to the dorsal commissure, also contained *gad-1*+ cells, probably marking pharyngeal neurons (figure 2r). The external opening of the pharynx, i.e. the mouth, contained a ring of *gad-1*+ cells that we refer to as the oral nerve ring (figure 2p). The schematics in figure 2 provide a summary of these results as would be observed in a transverse section through the anterior condensation (figure 2w,x).

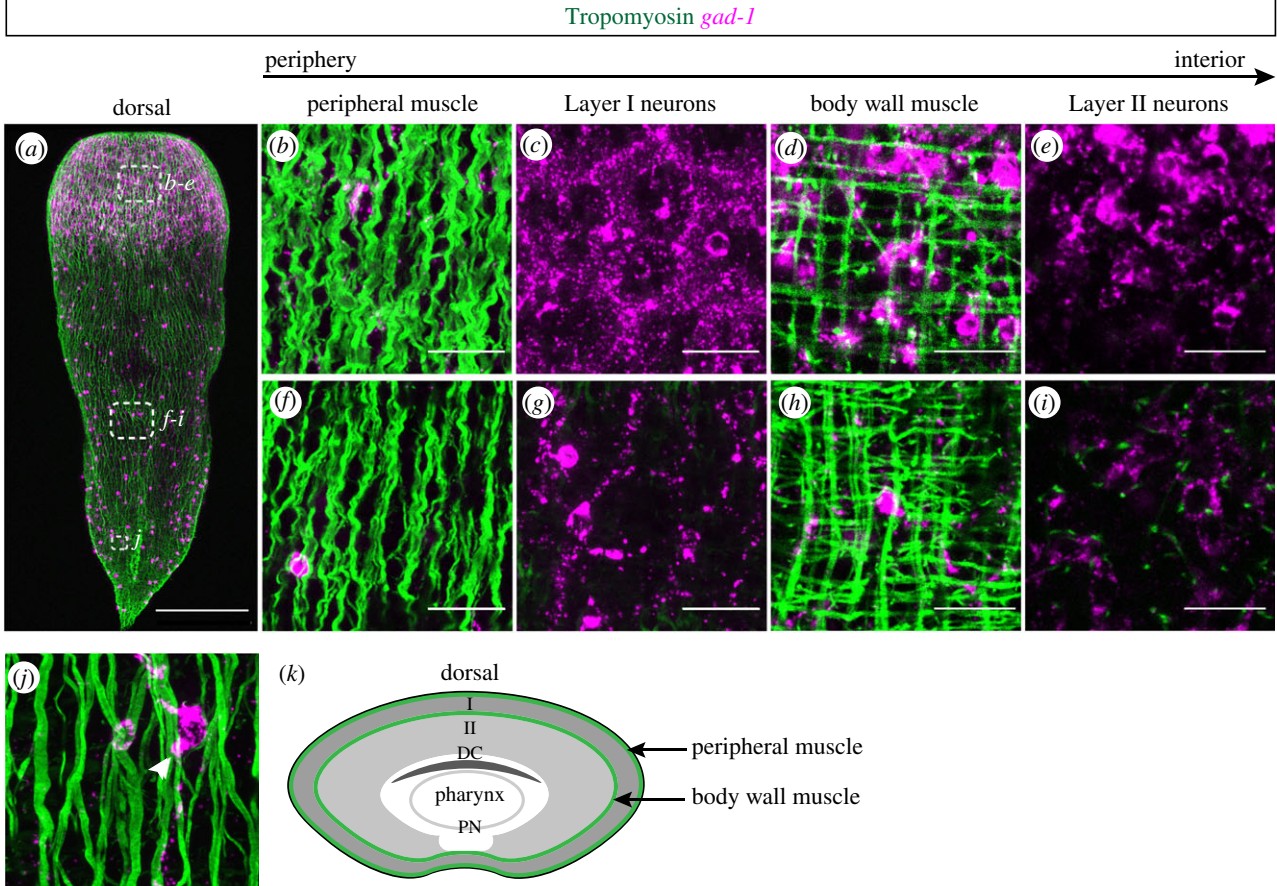

Tropomyosin *gad-1*

periphery → interior

dorsal | peripheral muscle | Layer I neurons | body wall muscle | Layer II neurons

**Figure 3.** Layer I neurons are located internal to peripheral musculature whereas Layer II neurons are internal to body wall musculature. (*a*) Dorsal projection of musculature (IHC using custom antibodies for Tropomyosin) and neurons (determined by *gad-1* expression). Dashed white boxes show regions imaged in *b–j*. (*b–e*) In the anterior, adjacent z-planes moving from periphery (left) of animal to the interior (right). (*b*) Peripheral muscle located external to all other structures. (*c*) Layer I neurons located internal to peripheral muscle. (*d*) Body wall muscle interacting with Layer I neurons, internal to the peripheral muscle. (*e*) Layer II neurons located internal to the body wall muscle. (*f–i*) In the posterior, adjacent z-planes moving from periphery (left) of animal to the interior (right). The spatial arrangement of the musculature and neurons found in the anterior is recapitulated in the posterior. (*i*) Sparse *gad-1*⁺ cells internal to the body wall muscle are found in close proximity to muscle fibres. (*j*) High magnification view showing neuronal cell bodies (white arrowhead) and axons in close alignment with peripheral muscle fibres. (*k*) A schematic transverse section through the anterior depicting the location of the peripheral muscle and body wall muscle cell layers (green lines) relative to *gad-1* neuroanatomy. Scale bars: (*a*) 200 µm, (*b–j*) 20 µm. (Online version in colour.)

Another gene in our FISH screen, *transient receptor potential ion channel M-1 (TrpM-1)*, labelled cells throughout the animal but its expression was concentrated in the anterior, revealing further structures within the anterior condensation (figure 2*t*). *TrpM-1*⁺ cells were arranged in clusters, reminiscent of the clusters of nuclei observed previously within the neurite bundles (figure 2*b–h*). Double-FISH with *TrpM-1* and *gad-1* probes revealed that *TrpM-1*⁺ cell clusters were surrounded by *gad-1*⁺ neurite bundles within Layer I (figure 2*t,u*). Imaging of these *TrpM-1*⁺ cell clusters in transverse view showed that they are present in both Layer I and Layer II, and that the clusters are contiguous between the two layers (figure 2*v*).

## (c) Nervous system components are present both internal and external to body wall musculature

Xenacoelomorph species display variable placement of neural elements relative to body wall musculature [33,40], e.g. intraepidermal nerve nets external to musculature (xenoturbellids), condensed neural elements placed external to body wall musculature (nemertodermatids), and neural condensations present internal to body wall musculature (acoels). In *H. miamia*, histological studies had revealed a subepidermal nerve plexus present external to longitudinal body wall

musculature, and some neural cells present internally, close to the statocyst [46,53]. To determine the spatial relationship of the musculature and the *H. miamia* nervous system using molecular markers, we coupled Tropomyosin immunostaining using custom antibodies with *gad-1* FISH (figure 3; electronic supplementary material, figure S2, and movies S1, S2). This identified two major muscular structures, peripheral longitudinal muscle and a set of internal body wall musculature that includes circumferential, longitudinal and diagonal muscle fibers. In the anterior, Layer I *gad-1*⁺ neurite bundles were found internal to the peripheral muscle but external to the body wall musculature (figure 3*a–d*), with Layer II *gad-1*⁺ cells lying internal to the body wall musculature (figure 3*d,e*). A similar organization was observed in more posterior regions of the animal with the *gad-1*⁺ nerve net sitting between the peripheral and body wall muscle layers (figure 3*a,f,g,h,i*). We also observed the presence of *gad-1*⁺ cells internal to the body wall muscle, which could not be distinguished from the nerve net without labelling of muscle (figure 3*i*). Further studies will be needed to determine if the close connections between muscle fibres and *gad-1*⁺ cells (figure 3*j*) represent neuromuscular junctions. The schematic in figure 3 provides an overview of neural components in relation to the body wall musculature (figure 3*k*).

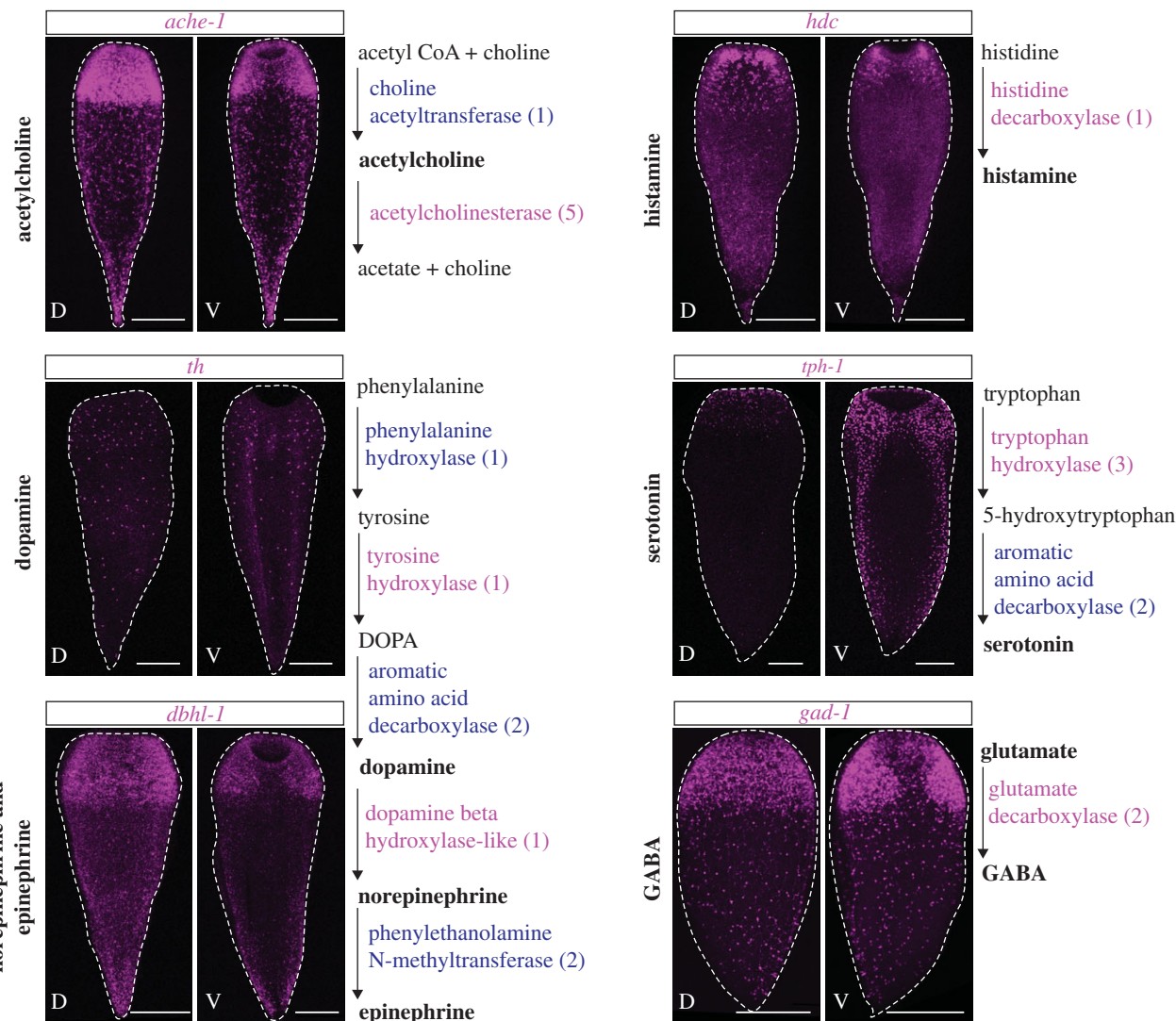

**Figure 4.** Neurotransmitter synthesis pathway gene expression labels cells throughout the animal, concentrated in the anterior. Major neurotransmitter synthesis pathways shown with enzymes highlighted in blue (orthologues encoded in the genome) and magenta (orthologues encoded in the genome and expression pattern shown) and neurotransmitters in bold. Numbers next to each enzyme represent the number of putative orthologues found in the *H. miamia* transcriptome (see electronic supplementary material, table S1). Paired images for each gene show dorsal projection (D) on left and ventral projection (V) on right. Dashed white lines around specimens show the outline of the animal. Scale bars 200 μm. (Online version in colour.)

## (d) Neurotransmitter synthesis genes are expressed in diverse neural populations

With a better understanding of *H. miamia* neuroanatomy and its genomic repertoire of neurotransmitter synthesis pathways, we sought to determine the number and spatial distribution of neural cell subtypes. Our efforts in determining the expression patterns of all identified homologues of neurotransmitter synthesis genes (electronic supplementary material, table S1) yielded expression patterns for at least one diagnostic enzyme per pathway (figure 4; electronic supplementary material, figure S1a). Notably, whereas all genes showed different numbers of labelled cells, these cells were distributed in a similar pattern along the AP axis. All genes examined labelled cells with a higher density in the anterior and more sparsely spaced distribution in the rest of the animal. This pattern mirrored the neural anatomy of *H. miamia* inferred by immunostaining and *gad-1* expression (figure 2).

The expression of markers of different neurotransmitters in the region of the anterior condensation could reflect three potential hypotheses. First, these markers might label distinct cell types that are present throughout the anterior condensation with no clear organization. Second, these markers might label distinct cell types that are localized to different regions of the anterior condensation. Finally, these markers may be expressed in the same cells, reflecting the presence of neurons with multiple molecular functionalities (i.e. cells that signal via more than one neurotransmitter). We used the layered structure of *gad-1*⁺ cells as a framework for studying the spatial organization of cells expressing the other neurotransmitter genes.

Cells expressing *tyrosine hydroxylase* (*th*) and *tryptophan hydroxylase-1* (*tph-1*) differed in spatial organization from *gad1*⁺ cells, labelling cells that were restricted to Layer I (figure 5). There was negligible co-expression of these two markers with *gad-1*, suggesting the presence of at least three different neural populations based on the neurotransmitter synthesis genes that are spatially restricted relative to each other (figure 5).

Double-FISH of *gad-1* with *acetylcholinesterase-1* (*ache-1*), *dopamine beta hydroxylase like-1* (*dbhl-1*), *histidine decarboxylase* (*hdc*) and *prohormone convertase* (*pc2*), a previously studied neural gene in *H. miamia*, showed that cells expressing these markers were present in Layer I and Layer II, but absent from the commissure, pharynx and oral nerve ring. Strikingly,

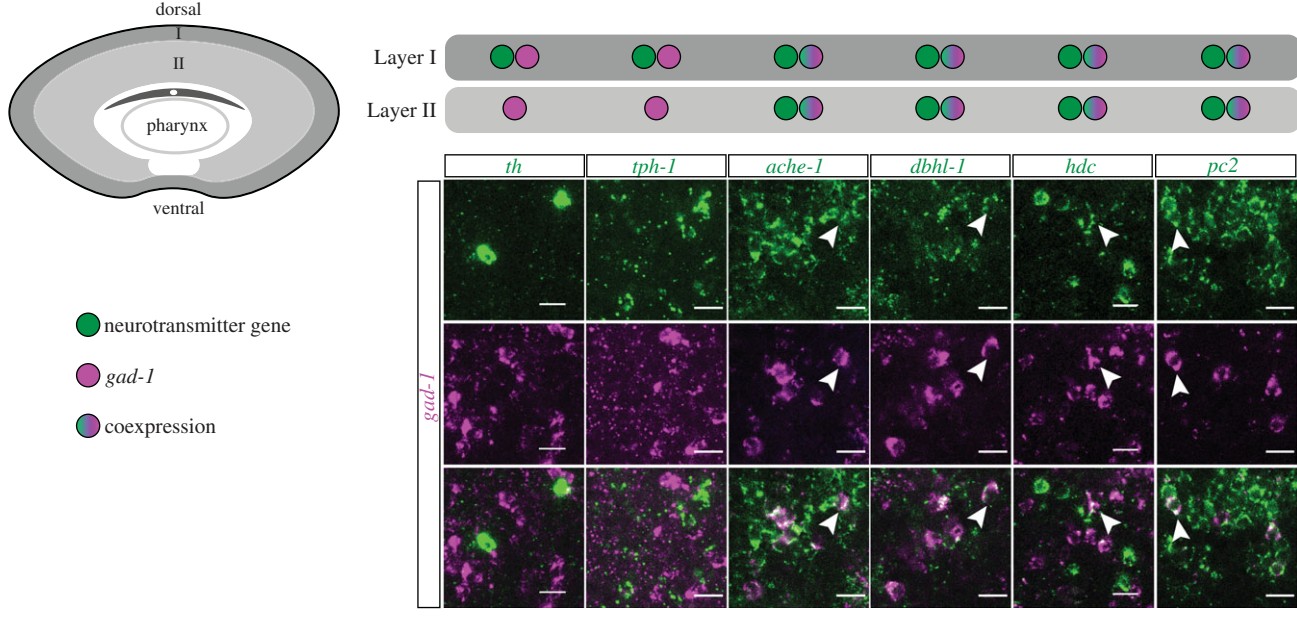

**Figure 5.** Distinct populations of neurons and multi-functional neurons are present in the two layers of the anterior condensation. Top left schematic depicts a transverse section through the anterior of the animal demarcating major neural structures, including Layer I and Layer II. To the right, co-expression data are summarized for Layer I (dark grey) and Layer II (light grey) as follows: the presence of cells with only neurotransmitter gene expression (green circle), only *gad-1* expression (magenta circle), or double-expression (magenta + green circle); legend on left. *th*⁺ cells and *tph-1*⁺ cells were only present in Layer I and showed very little to no co-expression with *gad-1*. By contrast, all *gad-1*⁺ neurons, in both Layer I and Layer II, were labelled with *ache-1*, *dbhl-1*, *hdc* (low levels) and *pc2*. White arrowheads indicate examples of double-labelled cells. Scale bars 20 µm. (Online version in colour.)

whereas many cells expressed only these markers, nearly all of the *gad-1*⁺ cells were double positives (figure 5), suggesting that putative gabaergic cells may also have cholinergic, adrenergic, histaminergic and peptidergic capacities in *H. miamia*. Further, *ache-1* and *dbhl-1* showed complete co-expression with *pc2* (electronic supplementary material, figure S3), indicating that the majority of neurons in *H. miamia* are peptidergic, cholinergic and adrenergic, with a subset of these having gabaergic and histaminergic functionalities as well.

## (e) *Hofstenia miamia* shows robust nervous system regeneration

With a clearer understanding of neural structures and cell types within *H. miamia*, we sought to determine the spatiotemporal nature of nervous system regeneration. Previous work has shown that *H. miamia* is capable of robust, whole-body regeneration [27,53]. Because regenerated worms show restoration of behaviours and can live and reproduce for many years following amputation, it is presumed that all neural cell types and structures return during this process. We therefore assessed if and when neural cell types and structures identified in this study are regenerated.

We used the expression of *gad-1* across a regenerative time course to determine the timeline for regeneration of major neural structures (figure 6). Animals were amputated transversely across the middle stripe, yielding head and tail fragments. By 9 h post amputation (hpa) and 15 hpa, there was no concentrated *gad-1* expression detectable at the wound site of regenerating tail fragments. Based on morphology and previous work [27], we found that both head and tail fragments had closed their wounds along the amputation site by 2 days post amputation (dpa). By 3 dpa, localized *gad-1* expression was detectable in the anterior of the regenerating tail fragments, indicative of newly differentiated *gad-1*⁺ neural cells. *gad-1* expression was expanded by

5 dpa and by 8 dpa, with neural structures including the ventral lobe-like structures formed by Layer II neurons and the dorsal commissure observable in the anterior of regenerating tail fragments. Thus, within 8 days, regenerating tail fragments recapitulated previously missing *gad-1*⁺ cells and structures and regained the ability to locomote (figure 6a). The timing of neural regeneration in *H. miamia* appears to be speedier than in another acoel, *Symsagittifera roscoffensis*, which regrows its head, brain and sensory organs within three to four weeks of decapitation [20,43,54]. Additionally, cells expressing other neural markers such as *dbhl-1*, *TrpC-1* and *pc2* also regenerated robustly (electronic supplementary material, figure S4). These findings, together with the return of normal movement, feeding, and reproductive behaviours seen in regenerated worms, suggest that *H. miamia* probably regenerates most, if not all, neural cell types.

In our studies of neural gene expression, the AP span of the anterior condensation appeared to scale in proportion to body length. To determine when the original ratio of anterior condensation span to body length is restored during regeneration, we measured the length of the anterior condensation, as labelled by *gad-1*, along the AP axis within regenerating fragments, relative to the body length of the entire fragment (figure 6b, electronic supplementary material, table S3). The anterior condensation length occupied a consistent proportion of the body length in intact animals, averaging to 26%. In 3 dpa regenerating tail fragments, when *gad-1* expression is first detectable in the anterior, it only occupied 8% of the length of the regenerating fragment. *gad-1* expression in regenerating tail fragments expanded and stabilized by 17 dpa, matching the 26% proportion seen in intact animals. In regenerating head fragments, the pre-existing anterior condensation occupied an outsized portion (51%) of the head fragment immediately after amputation (0 hpa). However, by 5 dpa, the size of the anterior condensation was reduced in regenerating head fragments to 26%. Thus, the anterior condensation

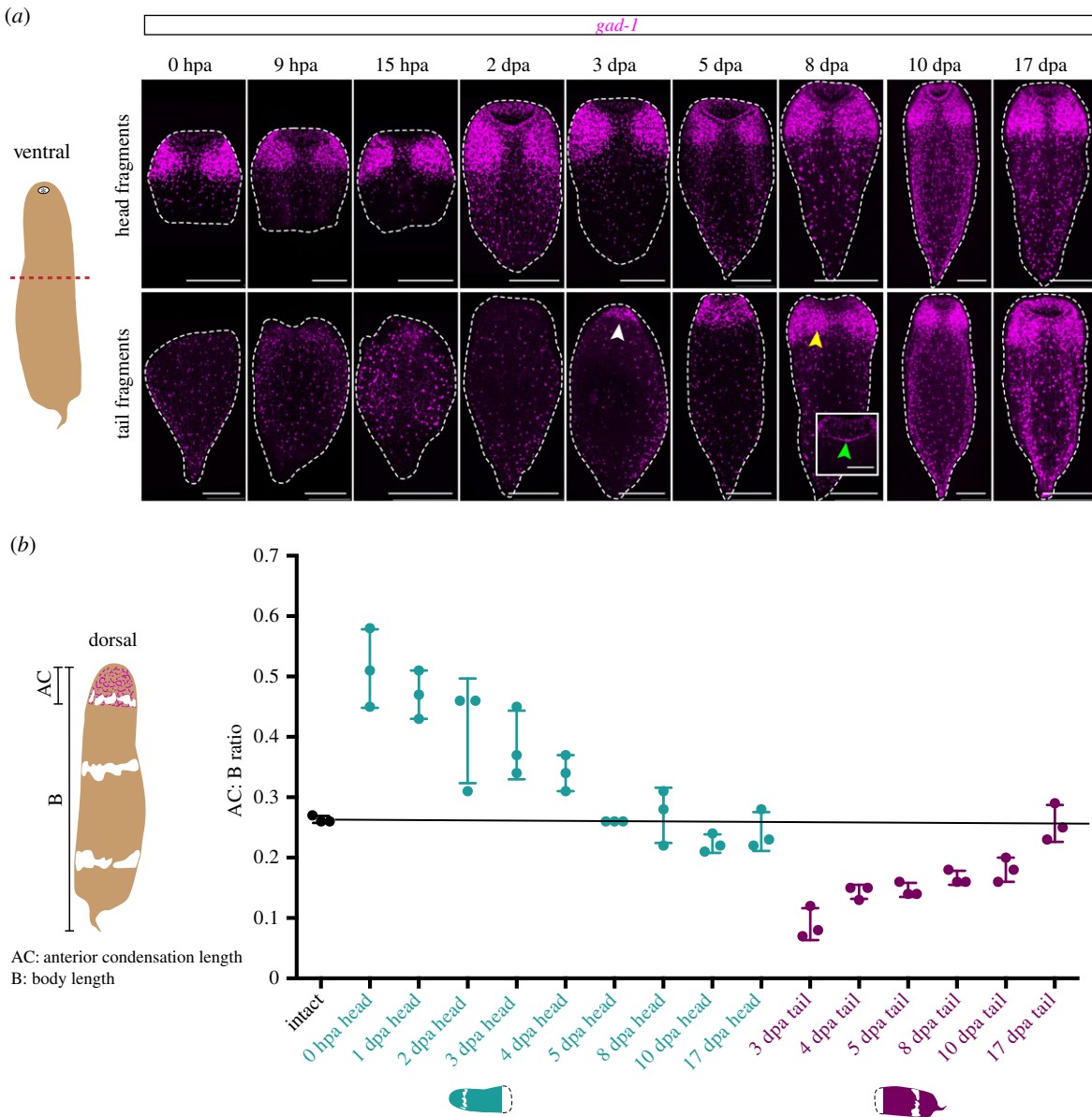

**Figure 6.** Regeneration and re-proportioning of the anterior condensation. (*a*) A schematic of *H. miamia* in ventral view (asterisk denotes the mouth) with amputation plane indicated as dashed red line. The amputation removed concentrated *gad-1* expression from tail fragments 0 h post amputation (hpa). The anterior condensation, observed as concentrated expression of *gad-1*, was detectable 3 days post amputation (dpa) in tail fragments (white arrowhead). The ventral lobe-like structures formed by Layer II neurons (yellow arrowhead). Inset shows a dorsal confocal z-plane in the anterior of the same animal, revealing that the dorsal commissure (green arrowhead) was regenerated by 8 dpa. Dashed white lines around specimens show the outline of the animal. (*b*) A schematic of *H. miamia* in dorsal view depicting *gad-1* anterior expression in magenta, with representative measurements for the anterior condensation length (AC) and body length (B). In intact animals (black), the anterior condensation occupied, on average, 26% of the body length of the animal. This ratio was restored in regenerating head fragments (teal) by 5 dpa and in tail fragments (purple) by 17 dpa. Three fragments were measured per time point. Scale bars 200 μm. (Online version in colour.)

## 4. Discussion

We investigated neural anatomy and neurotransmitter synthesis pathway genes in the acoel *H. miamia*, an early-diverging lineage within acoels [39,55]. In addition to a full complement of neurotransmitter synthesis enzymes encoded in the genome, we found a nerve net and an anterior condensation containing neurite bundles external to body wall musculature and a ring commissure and ventral lobe-like structures internal to the body wall musculature, which are features that had previously only been observed within in-group Crucimusculata acoels [19,22,36,40,50,55]. We hypothesize that these are putatively ancestral features of the acoel nervous system. However, studies comparing the mechanisms by which these structures develop and regenerate are needed to assess this proposed homology.

In addition to structures shared with other acoels, the *H. miamia* nervous system also contains unique features. Notably, we did not detect nerve cords in *H. miamia*, which could reflect the ancestral state among acoels. This supports the idea of independent evolution of nerve cords in xenacoelomorphs, which was suggested by recent work [37]. The anterior condensation of *H. miamia* exhibits a layered

becomes correctly proportioned to the size of the worm early upon amputation in regenerating head fragments (5 dpa) and later in tail fragments (17 dpa).

structure, which has not been described in other acoels. The reticulating neurite bundles in Layer I (outer layer) are reminiscent of structures observed in the *Symsagittifera roscoffensis* (Crucimusculata) brain [36,38,40,43,50,56,57], but higher-resolution studies are needed to assess if those neurite bundles also surround clustered nuclei with a putative sensory cell in the centre as they do in *H. miamia*. Furthermore, both layers of the *H. miamia* anterior condensation, excluding the ventral midline in Layer II, occupy the full span of the DV axis of the animal. This is in contrast with members of Crucimusculata, where this condensation (or brain) is located either centrally or dorsally when observed in transverse section [19,38,40]. Therefore, studies of how the anterior condensation is positioned in *H. miamia* could inform the evolution of mechanisms for localization of the central nervous system along the DV axis.

Our molecular characterization of neural cells in *H. miamia* defined distinct neuronal populations, but also revealed cells that have a putative capacity for functioning across multiple neurotransmitter families. For example, many cells expressed genes encoding enzymes for production of peptidergic, cholinergic and gabaergic signals (figure 5). Future studies will reveal whether this mRNA co-expression corresponds to co-existence, co-release and/or co-transmission of multiple neurotransmitters. The targets of these cells have yet to be identified, but this work can be facilitated via functional studies of the genes that encode neurotransmitter receptor proteins. *H. miamia* is an ideal research organism for functional studies of these questions, as it is amenable to robust and systemic RNAi.

Here, we provide an atlas for the regeneration of neural structures and cell types in *H. miamia*. This work will enable mechanistic studies of neural regeneration, including pathways that control the differentiation of stem cells into neural cells that integrate into a functional brain. Acoels represent a major but understudied phyletic lineage within the Bilateria, and therefore, the identification of transcriptional regulatory programs that enable the generation of diverse neuronal cell types during regeneration and development in *H. miamia* will inform our understanding of the evolution of nervous systems.

Ethics. This article does not present research with ethical considerations.

Data accessibility. The data and images are provided within the article and electronic supplementary material figures, tables and movies. Accession numbers for genes in this study are provided in electronic supplementary material, table S1.

Authors' contributions. R.E.H. and M.S. designed the study. R.E.H conducted the experiments and analyses and D.P. assisted. R.E.H. and M.S. wrote the manuscript.

Competing interests. We declare we do not have any competing interests.

Funding. M.S. was supported by the Searle Scholar Program and the Smith Family Foundation. R.E.H. was funded by the Department of Organismic and Evolutionary Biology at Harvard University. D.P. was supported by a Grant-in-aid of Undergraduate Research from the Museum of Comparative Zoology at Harvard University. This manuscript has been published by a grant from the Wetmore Colles Fund.

Acknowledgements. We thank Dr Lorenzo Ricci for guidance and help in generating antibodies and optimizing immunostaining. We would also like to thank Alyson Ramirez, Dr Lorenzo Ricci, Dr D. Marcela Bolanos, Amaneet Lochab, Dr Gonzalo Giribet, Dr Desmond Ramirez and Denise Li for their helpful comments on the manuscript.

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
