## [Reviewer comments · Proceedings of the Royal Society B: Biological Sciences]

Review History

RSPB-2020-0539.R0 (Original submission)

Review form: Reviewer 1 (Jose Martin-Duran)

Recommendation

Accept with minor revision (please list in comments)

Scientific importance: Is the manuscript an original and important contribution to its field?

Excellent

General interest: Is the paper of sufficient general interest?

Excellent

Quality of the paper: Is the overall quality of the paper suitable?

Excellent

Is the length of the paper justified?

Yes

Should the paper be seen by a specialist statistical reviewer?

No

Do you have any concerns about statistical analyses in this paper? If so, please specify them explicitly in your report.

No

It is a condition of publication that authors make their supporting data, code and materials available - either as supplementary material or hosted in an external repository. Please rate, if applicable, the supporting data on the following criteria.

Is it accessible?

Yes

Is it clear?

Yes

Is it adequate?

Yes

Do you have any ethical concerns with this paper?

No

Comments to the Author

In "Neural architecture and regeneration in the acoel *Hofstenia miamia*", Hulett and co-workers describe morphologically and molecularly the neural architecture and its regeneration in a representative of Acoelomorpha (*Hofstenia miamia*) that is emerging as the main research model species of this group. Combining immunoreactivity patterns against FMRF-amide, Tyrosinated Tubulin and tropomyosin with two neural gene markers, the authors demonstrate that the nervous system of *Hofstenia* comprises an anterior condensation organised in two distinct layers and lack condensed nerve cords. Unlike other Xenacoelomorphs, *Hofstenia* shows a complete repertoire of neurotransmitters, and the authors report the localisation and co-expression of core representative genes of this repertoire. Finally, they use these molecular tools to describe anterior and posterior neural regeneration. The manuscript is a joy to read and the figures are clear, informative and appealing. This work is an important step to better understand a key animal lineage in evolutionary developmental biology, and I am convinced it will form the basis for future studies in evolutionary and regenerative biology of the nervous system in this species.

I do not have major concerns and only a handful of minor comments or suggestions:

- Line 187-192: As I understand them, I find these two sentences a bit odd, because the position of acoels within Bilateria has never been questioned. I would perhaps rephrase them and just focus on the neurotransmitters (acoel data supports that most were in Cnidaria+Bilateria, but histamine is a Bilateria innovation).

- The authors might want to clarify why they chose glutamate decarboxylase (*gad-1*) and transient receptor potential ion channel M (*TrpM-1*) as gene markers to characterise nervous system structure, instead of perhaps other more classical pan-neural bilaterian markers (e.g. synapsin, *elav*)

- Line 251: It might be good to clarify why the relationship between the nervous system and the musculature matters.

- Line 339-341: the sentence reads to me as if because *Hofstenia* is an early-divergent lineage, what it shows is the ancestral condition, which is not necessarily true (the ancestral trait will be inferred from outgroup comparison). I would clarify that point, which is very clear from Figure 1 (outgroup comparison with Bilaterians and Cnidarians supports *Hofstenia* showing the most likely ancestral state and probably highlight again [after what is said in the Results, line 193-196] that e.g. *Xenoturbella* has had simplification at the level of neurotransmitters and additional

secondary independent losses of particular neurotransmitters in different acoel species, such as GABA or dopamine)

Review form: Reviewer 2

Recommendation

Major revision is needed (please make suggestions in comments)

Scientific importance: Is the manuscript an original and important contribution to its field?

Good

General interest: Is the paper of sufficient general interest?

Acceptable

Quality of the paper: Is the overall quality of the paper suitable?

Good

Is the length of the paper justified?

Yes

Should the paper be seen by a specialist statistical reviewer?

No

Do you have any concerns about statistical analyses in this paper? If so, please specify them explicitly in your report.

No

It is a condition of publication that authors make their supporting data, code and materials available - either as supplementary material or hosted in an external repository. Please rate, if applicable, the supporting data on the following criteria.

Is it accessible?

No

Is it clear?

N/A

Is it adequate?

Yes

Do you have any ethical concerns with this paper?

No

Comments to the Author

The manuscript "Neural architecture and regeneration in the acoel *Hofstenia miamia*" by Hulett et al provides a description of the adult nervous system morphology, neurotransmitter pathway expression and regeneration experiments in the acoel *Hofstenia miamia*. The manuscript aims to deliver a foundation for further studies of regeneration in this species.

Overall the manuscript is concise, well references and well balanced in the way it relates the results to the partly disputed phylogenetic position of acoels. I have several points of improvement of the manuscript, most of them are minor.

Major point:

The authors state at several places in the manuscript that they provide the study of the

regeneration of neural cell types. However, the manuscript is missing the expression of the markers used to identify the cell types in the regeneration experiments. Only the expression of *gad-1* is provided and it therefore remains unclear if all neural cell types, or only a subset is regenerated. A double FISH with *gad-1* and the markers (as in figure 5), would complete the picture of the regeneration including the sequence of cell type regeneration. Because the probes seem to be synthesized and the fragments likely preserved and available, adding these descriptions should not be difficult and doable in an extended time of a major revision of the manuscript.

Minor points:

- Steinböck conducted regeneration experiments in *Hofstenia giselae* (<https://doi.org/10.1007/BF01380539>). Although it is a different species, how do the presented experiments and the regeneration in general compare with *H. miamia*?

The description of how the animals are kept is a bit short and lacks specifics. Is there maybe a reference to a more elaborated description of the culture? Also, please list here the full species name including the descriptor.

Not sure why horse-serum was used for the blocking, when the secondary anti-bodies were raised in goat? I am curious about the reasoning.

Line 104: Please rephrase this sentence to make it clearer: "Neural cells were only present internal to the peripheral longitudinal muscle, but were found on either side of the body wall musculature."

Is the Statocyst regenerated?

Line 204: "Although the anterior structure meets the generally accepted definition of the term "brain", which has been applied to anterior condensations in other acoels, we refrain from using this term to avoid an implication of homology with the brains of other xenacoelomorphs or other bilaterians. In this manuscript, we refer to this structure with its many organized regions as the anterior condensation." I disagree in this point with the authors and would like to encourage them to revise their decision. The use of a term does not and should not imply a homology proposition per se. Many terms such as larva, nerve cords etc. are descriptive terms, which, when properly described, are useful. There is no way around re-using the term for convergent structures. The bird wing is still a wing as it is the *Drosophila* wing. Otherwise we would have to find a new term for the one or the other. I suggest to maybe reference the definition of a brain here – or shortly describe what the authors mean.

Can the confocal stacks be made accessible in a public depository after publication? It would be helpful to also gain a better impression of the muscle/neural layers in the body wall. Last could in addition be added as supplement movie to the paper.

Line 308: "However, whether all diverse neural cell types and structures identified in this study are regenerated remains to be determined, as does the timing of their regeneration."

Line 338: "neural components encoded in the genome". Well, there are a multitude of neural components. Please restrict this sentence to the neurotransmitter pathway here.

Line 340: "our work suggests that the last common ancestral acoel had a more complex nervous system, molecularly and structurally, than has previously been hypothesized." I can not follow this statement because I think it is not justified from the data provided. An anterior brain was always proposed as ancestral state for Acoela and the nerve plexus was described before and dates back to the times of Steinböck and earlier. Please support this statement with references if I am wrong.

- Figure 2: Is it possible to use the same color for the same marker. Tyr-tub is green in 2g,h and magenta in i-k.
- Figure 2: What is the posterior signal in 2b?
- Figure 2: Can the muscle layer be indicated in 2x?
- Figure 3: Can the authors add a lateral view of the confocal stack to illustrate the layers?
- Figure 3: What is outside the peripheral muscle layer? How is the epidermis structured?

Schemes in Figure 2, 3, 5 and Supp. Figure 2 : These figures are a bit too schematic. It would be helpful to outline the cells in these crosssections, maybe with help of published histological sections, the confocal stacks etc. It is hard to understand how many cells form one layer, how the epidermis is composed and what the filling tissue (parenchyme?) is composed of. I suggest to elaborate this scheme a bit more.

Decision letter (RSPB-2020-0539.R0)

17-Apr-2020

Dear Dr Srivastava,

We have now received referees' reports on your manuscript RSPB-2020-0539 entitled "Neural architecture and regeneration in the acoel *Hofstenia miamia*".

Whilst both referees are generally positive about the paper, they have recommended several revisions, some of which are substantial. The paper has therefore been rejected in its current form, but we would be happy to consider a resubmission, provided the comments of the referees are fully addressed. However please note that this is not a provisional acceptance.

To upload a resubmitted manuscript, log into <http://mc.manuscriptcentral.com/prsb> and enter your Author Centre, where you will find your manuscript title listed under "Manuscripts with Decisions." Under "Actions," click on "Create a Resubmission". Please be sure to indicate in your cover letter that it is a resubmission, and supply the previous reference number.

Finally, I hope you and your co-authors are well in this challenging time.

Yours sincerely,

Professor Loeske Kruuk

Associate Editor

Board Member: 1

Comments to Author:

Two experts in the field have reviewed your manuscript and both identified some issues with it (both minor and major). Considering the reviewers' comments, I cannot recommend the manuscript in its current status for publication on Proc of the Royal Society B.

Reviewer(s)' Comments to Author:

Referee: 1

Comments to the Author(s)

In "Neural architecture and regeneration in the acoel *Hofstenia miamia*", Hulett and co-workers describe morphologically and molecularly the neural architecture and its regeneration in a representative of Acoelomorpha (*Hofstenia miamia*) that is emerging as the main research model species of this group. Combining immunoreactivity patterns against FMRF-amide, Tyrosinated Tubulin and tropomyosin with two neural gene markers, the authors demonstrate that the nervous system of *Hofstenia* comprises an anterior condensation organised in two distinct layers and lack condensed nerve cords. Unlike other Xenacoelomorphs, *Hofstenia* shows a complete repertoire of neurotransmitters, and the authors report the localisation and co-expression of core representative genes of this repertoire. Finally, they use these molecular tools to describe anterior and posterior neural regeneration. The manuscript is a joy to read and the figures are clear, informative and appealing. This work is an important step to better understand a key animal lineage in evolutionary developmental biology, and I am convinced it will form the basis for future studies in evolutionary and regenerative biology of the nervous system in this species.

I do not have major concerns and only a handful of minor comments or suggestions:

- Line 187-192: As I understand them, I find these two sentences a bit odd, because the position of acoels within Bilateria has never been questioned. I would perhaps rephrase them and just focus on the neurotransmitters (acoel data supports that most were in Cnidaria+Bilateria, but histamine is a Bilateria innovation).
- The authors might want to clarify why they chose glutamate decarboxylase (*gad-1*) and transient receptor potential ion channel M (*TrpM-1*) as gene markers to characterise nervous system structure, instead of perhaps other more classical pan-neural bilaterian markers (e.g. *synapsin*, *elav*)
- Line 251: It might be good to clarify why the relationship between the nervous system and the musculature matters.
- Line 339-341: the sentence reads to me as if because *Hofstenia* is an early-divergent lineage, what it shows is the ancestral condition, which is not necessarily true (the ancestral trait will be inferred from outgroup comparison). I would clarify that point, which is very clear from Figure 1 (outgroup comparison with Bilaterians and Cnidarians supports *Hofstenia* showing the most likely ancestral state and probably highlight again [after what is said in the Results, line 193-196] that e.g. *Xenoturbella* has had simplification at the level of neurotransmitters and additional secondary independent losses of particular neurotransmitters in different acoel species, such as GABA or dopamine)

Referee: 2

Comments to the Author(s)

The manuscript "Neural architecture and regeneration in the acoel *Hofstenia miamia*" by Hulett et al provides a description of the adult nervous system morphology, neurotransmitter pathway expression and regeneration experiments in the acoel *Hofstenia miamia*. The manuscript aims to deliver a foundation for further studies of regeneration in this species.

Overall the manuscript is concise, well references and well balanced in the way it relates the results to the partly disputed phylogenetic position of acoels. I have several points of improvement of the manuscript, most of them are minor.

Major point:

The authors state at several places in the manuscript that they provide the study of the regeneration of neural cell types. However, the manuscript is missing the expression of the markers used to identify the cell types in the regeneration experiments. Only the expression of *gad-1* is provided and it therefore remains unclear if all neural cell types, or only a subset is regenerated. A double FISH with *gad-1* and the markers (as in figure 5), would complete the picture of the regeneration including the sequence of cell type regeneration. Because the probes seem to be synthesized and the fragments likely preserved and available, adding these descriptions should not be difficult and doable in an extended time of a major revision of the manuscript.

Minor points:

- Steinböck conducted regeneration experiments in *Hofstenia giselae* (<https://doi.org/10.1007/BF01380539>). Although it is a different species, how do the presented experiments and the regeneration in general compare with *H. miamia*?

The description of how the animals are kept is a bit short and lacks specifics. Is there maybe a reference to a more elaborated description of the culture? Also, please list here the full species name including the descriptor.

Not sure why horse-serum was used for the blocking, when the secondary anti-bodies were raised in goat? I am curious about the reasoning.

Line 104: Please rephrase this sentence to make it clearer: "Neural cells were only present internal to the peripheral longitudinal muscle, but were found on either side of the body wall musculature."

Is the Statocyst regenerated?

Line 204: "Although the anterior structure meets the generally accepted definition of the term "brain", which has been applied to anterior condensations in other acoels, we refrain from using this term to avoid an implication of homology with the brains of other xenacoelomorphs or other bilaterians. In this manuscript, we refer to this structure with its many organized regions as the anterior condensation." I disagree in this point with the authors and would like to encourage them to revise their decision. The use of a term does not and should not imply a homology proposition per se. Many terms such as larva, nerve cords etc. are descriptive terms, which, when properly described, are useful. There is no way around re-using the term for convergent structures. The bird wing is still a wing as it is the *Drosophila* wing. Otherwise we would have to find a new term for the one or the other. I suggest to maybe reference the definition of a brain here - or shortly describe what the authors mean.

Can the confocal stacks be made accessible in a public depository after publication? It would be helpful to also gain a better impression of the muscle/neural layers in the body wall. Last could in addition be added as supplement movie to the paper.

Line 308: "However, whether all diverse neural cell types and structures identified in this study are regenerated remains to be determined, as does the timing of their regeneration."

Line 338: "neural components encoded in the genome". Well, there are a multitude of neural components. Please restrict this sentence to the neurotransmitter pathway here.

Line 340: "our work suggests that the last common ancestral acoel had a more complex nervous system, molecularly and structurally, than has previously been hypothesized." I can not follow this statement because I think it is not justified from the data provided. An anterior brain was always proposed as ancestral state for Acoela and the nerve plexus was described before and dates back to the times of Steinböck and earlier. Please support this statement with references if I am wrong.

- Figure 2: Is it possible to use the same color for the same marker. Tyr-tub is green in 2g,h and magenta in i-k.
- Figure 2: What is the posterior signal in 2b?
- Figure 2: Can the muscle layer be indicated in 2x?
- Figure 3: Can the authors add a lateral view of the confocal stack to illustrate the layers?
- Figure 3: What is outside the peripheral muscle layer? How is the epidermis structured?

Schemes in Figure 2, 3, 5 and Supp. Figure 2: These figures are a bit too schematic. It would be helpful to outline the cells in these crosssections, maybe with help of published histological sections, the confocal stacks etc. It is hard to understand how many cells form one layer, how the epidermis is composed and what the filling tissue (parenchyme?) is composed of. I suggest to elaborate this scheme a bit more.

Author's Response to Decision Letter for (RSPB-2020-0539.R0)

See Appendix A.

RSPB-2020-1198.R0

Review form: Reviewer 2

Recommendation

Accept as is

Scientific importance: Is the manuscript an original and important contribution to its field?

Good

General interest: Is the paper of sufficient general interest?

Acceptable

Quality of the paper: Is the overall quality of the paper suitable?

Good

Is the length of the paper justified?

Yes

Should the paper be seen by a specialist statistical reviewer?

No

Do you have any concerns about statistical analyses in this paper? If so, please specify them explicitly in your report.

No

It is a condition of publication that authors make their supporting data, code and materials available - either as supplementary material or hosted in an external repository. Please rate, if applicable, the supporting data on the following criteria.

Is it accessible?

N/A

Is it clear?

Yes

Is it adequate?

Yes

Do you have any ethical concerns with this paper?

No

Comments to the Author

The authors have addressed all my comments satisfactory. However, if the lab opens during the manuscript is in production, it would be good to add the co-expression studies, since the authors "would have been happy to perform the co-expression studies recommended by the reviewer". It's good when the authors are happy.

Decision letter (RSPB-2020-1198.R0)

17-Jun-2020

Dear Dr Srivastava

I am pleased to inform you that your Review manuscript RSPB-2020-1198 entitled "Neural architecture and regeneration in the acoel *Hofstenia miamia*" has been accepted for publication in Proceedings B.

The Associate Editor has not recommend any further changes. The referee comments that adding the co-expression studies would be ideal if feasible, but I am assuming this would not be possible and therefore am happy to accept your paper as it is. Please proof-read your manuscript carefully and upload your final files for publication. Because the schedule for publication is very tight, it is a condition of publication that you submit the revised version of your manuscript within 7 days. If you do not think you will be able to meet this date please let me know immediately.

To upload your manuscript, log into <http://mc.manuscriptcentral.com/prsb> and enter your Author Centre, where you will find your manuscript title listed under "Manuscripts with Decisions." Under "Actions," click on "Create a Revision." Your manuscript number has been appended to denote a revision.

You will be unable to make your revisions on the originally submitted version of the manuscript. Instead, upload a new version through your Author Centre.

1) A text file of the manuscript (doc, txt, rtf or tex), including the references, tables (including captions) and figure captions. Please remove any tracked changes from the text before submission. PDF files are not an accepted format for the "Main Document".

2) A separate electronic file of each figure (tiff, EPS or print-quality PDF preferred). The format should be produced directly from original creation package, or original software format. Please note that PowerPoint files are not accepted.

3) Electronic supplementary material: this should be contained in a separate file from the main text and the file name should contain the author's name and journal name, e.g. `authorname_procb_ESM_figures.pdf`

All supplementary materials accompanying an accepted article will be treated as in their final form. They will be published alongside the paper on the journal website and posted on the online figshare repository. Files on figshare will be made available approximately one week before the accompanying article so that the supplementary material can be attributed a unique DOI. Please see: <https://royalsociety.org/journals/authors/author-guidelines/>

4) Data-Sharing and data citation

It is a condition of publication that data supporting your paper are made available. Data should be made available either in the electronic supplementary material or through an appropriate repository. Details of how to access data should be included in your paper. Please see <https://royalsociety.org/journals/ethics-policies/data-sharing-mining/> for more details.

<http://datadryad.org/submit?journalID=RSPB&manu=RSPB-2020-1198> which will take you to your unique entry in the Dryad repository.

Once again, thank you for submitting your manuscript to Proceedings B and I look forward to receiving your final version. If you have any questions at all, please do not hesitate to get in touch.

Finally, I hope you and your co-authors are well in these difficult times.

Yours sincerely,
Professor Loeske Kruuk
<mailto:proceedingsb@royalsociety.org>

Associate Editor
Board Member
Comments to Author:
Dear Dr Srivastava,

Your manuscript has been reviewed by an expert in the field and considering their comments and the broad interest of the work I am delighted to recommend your work for publication.

Best wishes,
Roberto

Reviewer(s)' Comments to Author:

Referee: 2

Comments to the Author(s).

The authors have addressed all my comments satisfactory. However, if the lab opens during the manuscript is in production, it would be good to add the co-expression studies, since the authors "would have been happy to perform the co-expression studies recommended by the reviewer". It's good when the authors are happy.

Decision letter (RSPB-2020-1198.R1)

29-Jun-2020

Dear Dr Srivastava

I am pleased to inform you that your manuscript entitled "Neural architecture and regeneration in the acoel *Hofstenia miamia*" has been accepted for publication in Proceedings B.

Open Access

Paper charges

Sincerely,
Proceedings B
<mailto:proceedingsb@royalsociety.org>

Appendix A

We thank the reviewers for their careful reading of the paper and for recognizing the value of this work. Their comments have helped us improve the manuscript. We explain below in detail how we have addressed most of the comments of the reviewers.

Reviewer(s)' Comments to Author:

Referee: 1

Comments to the Author(s)

In "Neural architecture and regeneration in the acoel *Hofstenia miamia*", Hulett and co-workers describe morphologically and molecularly the neural architecture and its regeneration in a representative of Acoelomorpha (*Hofstenia miamia*) that is emerging as the main research model species of this group. Combining immunoreactivity patterns against FMRF-amide, Tyrosinated Tubulin and tropomyosin with two neural gene markers, the authors demonstrate that the nervous system of *Hofstenia* comprises an anterior condensation organised in two distinct layers and lack condensed nerve cords. Unlike other Xenacoelomorphs, *Hofstenia* shows a complete repertoire of neurotransmitters, and the authors report the localisation and co-expression of core representative genes of this repertoire. Finally, they use these molecular tools to describe anterior and posterior neural regeneration. The manuscript is a joy to read and the figures are clear, informative and appealing. This work is an important step to better understand a key animal lineage in evolutionary developmental biology, and I am convinced it will form the basis for future studies in evolutionary and regenerative biology of the nervous system in this species.

Thank you! We appreciate your thorough reading of the manuscript and the important points you made below.

I do not have major concerns and only a handful of minor comments or suggestions:

- Line 187-192: As I understand them, I find these two sentences a bit odd, because the position of acoels within Bilateria has never been questioned. I would perhaps rephrase them and just focus on the neurotransmitters (acoel data supports that most were in Cnidaria+Bilateria, but histamine is a Bilateria innovation).

We thank the reviewer for catching our odd phrasing of this sentence. We did not mean to imply that these data support or have any bearing on the phylogenetic placement of acoels. Our objective was to convey that it is "no surprise" that many of these components were found in *H. miamia*. We have re-written the sentence as recommended.

The previous "These findings are consistent with the placement of *H. miamia* within the Bilateria, as many of these neural components are found in nephrozoans and in cnidarians, the bilaterian outgroup lineage, making them likely ancestral eumetazoan characteristics. Notably, the histamine synthesis pathway was found in the *H. miamia* genome but has not been detected in non-bilaterian genomes studied thus far, suggesting it is an ancestral bilaterian trait." now

reads “Many of these neural components are found in nephrozoans and in cnidarians, the outgroup lineage to bilaterians, and their presence in *H. miamia* are consistent with them being ancestral eumetazoan *sensu stricto* (cnidarian + bilaterian) characteristics. Notably, the histamine synthesis pathway was found in the *H. miamia* genome but has not been detected in non-bilaterian genomes studied thus far, suggesting it is a bilaterian innovation.”

- The authors might want to clarify why they chose glutamate decarboxylase (*gad-1*) and transient receptor potential ion channel M (*TrpM-1*) as gene markers to characterise nervous system structure, instead of perhaps other more classical pan-neural bilaterian markers (e.g. *synapsin*, *elav*)

Two putatively pan-neural markers, *pc2* and *synapsin*, had been studied previously in *H. miamia* (Srivastava M, Mazza-Curll KL, Van Wolfswinkel JC, Reddien PW. 2014 Whole-body acoel regeneration is controlled by Wnt and Bmp-Admp signaling. *Curr. Biol.* **24**, 1107–1113). The *synapsin* probe did not show strong labeling. Neither of these markers showed labeling of axons, neurite bundles, or other neural structures. In our screen of neurotransmitter synthesis enzymes, *gad-1* stood out as unique in that its mRNA can be detected in neural processes and reveals axons in the body, neurite bundles in the anterior condensation, and the dorsal commissure. Thus, this gene served as a good marker for orienting the community to major structures in the *H. miamia* nervous system. Similarly, in our screen of other neural marker genes, *TrpM-1* stood out as a unique marker for a cell population that complemented the pattern of *gad-1*, showing the presence of putative neural cells surrounded by *gad-1+* neurite bundles. We have modified the language in the paragraph that introduces *gad-1* expression (third paragraph in section (b) of Results).

- Line 251: It might be good to clarify why the relationship between the nervous system and the musculature matters.

Previous literature has paid much attention to the placement of the nerve net and condensed elements of the nervous system relative to body wall musculature in xenacoelomorph species (Gavilán B, Perea-Atienza E, Martínez P. 2016 Xenacoelomorpha: a case of independent nervous system centralization? *Philos. Trans. R. Soc. Lond. B Biol. Sci.* **371**; Raikova OI, Meyer-Wachsmuth I, Jondelius U. 2016 The plastic nervous system of Nemertodermatida. *Organisms Diversity & Evolution.* **16**, 85–104). Xenoturbellid nerve nets are intraepidermal, always placed external to body wall musculature. Most nemertodermatids have their nervous system external to the body wall musculature. In contrast, acoel “brains” tend to be internal to the body wall musculature. Therefore, it was important for us to investigate the placement of neural elements in *Hofstenia* relative to its musculature. To make our work relevant to this existing literature, we have added a sentence at the beginning of section (c) in Results. “The complement of neural genes shared between *H. miamia* and other bilaterians suggests that the lack of certain neurotransmitter synthesis pathways in xenoturbellids, nemertodermatids, and other acoels [26,32,45–49]”

- Line 339-341: the sentence reads to me as if because *Hofstenia* is an early-divergent lineage, what it shows is the ancestral condition, which is not necessarily true (the ancestral trait will be

inferred from outgroup comparison). I would clarify that point, which is very clear from Figure 1 (outgroup comparison with Bilaterians and Cnidarians supports Hofstenia showing the most likely ancestral state and probably highlight again [after what is said in the Results, line 193-196] that e.g. Xenoturbella has had simplification at the level of neurotransmitters and additional secondary independent losses of particular neurotransmitters in different acoel species, such as GABA or dopamine)

Thank you again for catching awkward wording. We absolutely agree that traits in early diverging lineages should not be conflated with ancestral traits. In view of this comment, and also comments by Reviewer 2, we have edited this section of the paper (first paragraph of discussion).

Referee: 2

Comments to the Author(s)

The manuscript “Neural architecture and regeneration in the acoel Hofstenia miamia” by Hulett et al provides a description of the adult nervous system morphology, neurotransmitter pathway expression and regeneration experiments in the acoel Hofstenia miamia. The manuscript aims to deliver a foundation for further studies of regeneration in this species.

Overall the manuscript is concise, well references and well balanced in the way it relates the results to the partly disputed phylogenetic position of acoels. I have several points of improvement of the manuscript, most of them are minor.

We thank this reviewer for challenging us to provide better justifications for some of our statements. In particular, we enjoyed revisiting the long-term debate we’ve had in our group and with some of our colleagues about the term “brain”.

Major point:

The authors state at several places in the manuscript that they provide the study of the regeneration of neural cell types. However, the manuscript is missing the expression of the markers used to identify the cell types in the regeneration experiments. Only the expression of *gad-1* is provided and it therefore remains unclear if all neural cell types, or only a subset is regenerated. A double FISH with *gad-1* and the markers (as in figure 5), would complete the picture of the regeneration including the sequence of cell type regeneration. Because the probes seem to be synthesized and the fragments likely preserved and available, adding these descriptions should not be difficult and doable in an extended time of a major revision of the manuscript.

We have now added data showing that the timeline of regeneration of *TrpC-1+* and *dbhl-1+* neurons is similar to the one shown for *gad-1* in figure 6 (new data in Supplementary figure 4). We had focused on the *gad-1* marker because it is unique in revealing multiple structures of the anterior condensation, such as the dorsal commissure, which are not observable with the other markers. Further, we added a day 8 regenerating tail fragment showing regeneration of *pc2+* neurons. The regeneration time course of these additional markers, and the fact that regenerated animals display normal behaviors (feeding, swimming) and go on to live for years,

support the idea that *Hofstenia* undergoes complete regeneration of its nervous system. We have added a sentence summarizing these results to the second paragraph of Results Section (e) and edited our use of the term “cell type” to avoid any suggestion that we have looked at all cell types.

We would have been happy to perform the co-expression studies recommended by the reviewer, however, our lab is shut down because of the pandemic and we will likely have very limited access over the next several months. We think the results in this manuscript are robust, and hope that the reviewer agrees that it is more advantageous for our research community to see this work rather than hold it back indefinitely.

Minor points:

- Steinböck conducted regeneration experiments in *Hofstenia giselae* (<https://doi.org/10.1007/BF01380539>). Although it is a different species, how do the presented experiments and the regeneration in general compare with *H. miamia*? Based on morphological studies and molecular phylogenetics, *H. giselae* is a junior synonym for *H. miamia* (Hooge M, Wallberg A, Todt C, Maloy A, Jondelius U, Tyler S. 2007 A revision of the systematics of panther worms (*Hofstenia* spp., Acoela), with notes on color variation and genetic variation within the genus. *Hydrobiologia* **592**, 439–454). Steinbock focused on the regenerated epidermis in amputated fragments, and based on our limited translations of the german text, we did not find substantial statements made about neural regeneration in his work. For example, Steinbock mentions being unable to ascertain the formation of the nerve fiber mass/nerveplexus because of the extremely tender/fragile samples (see pages 415-417 in Steinbock, O. 1967 Regenerationsversuche mit *Hofstenia giselae* Steinb. (*Turbellaria acoela*). *W. Roux' Archiv f. Entwicklungsmechanik* **158**, 394-458).

The description of how the animals are kept is a bit short and lacks specifics. Is there maybe a reference to a more elaborated description of the culture? Also, please list here the full species name including the descriptor.

We have added a reference to the original paper where the model system was first described (Srivastava M, Mazza-Curll KL, Van Wolfswinkel JC, Reddien PW. 2014 Whole-body acoel regeneration is controlled by Wnt and Bmp-Admp signaling. *Curr. Biol.* **24**, 1107–1113).

Not sure why horse-serum was used for the blocking, when the secondary anti-bodies were raised in goat? I am curious about the reasoning.

Horse serum is used as a blocking agent in many standard immunostaining protocols, much like western blocking reagent. See for example the product page (<https://www.sigmaaldrich.com/catalog/product/sigma/h0146?lang=en®ion=US>). Further, numerous papers have been published using this in our field, particularly in planarian worms (see for example: Pascolini R, Rosa ID, Fagotti A, Panara F, Gabbiani G. 1992 The mammalian anti- α -smooth muscle actin monoclonal antibody recognizes an α -actin-like protein in planaria (*Dugesia lugubris* s.l.). *Differentiation* **51**, 177–186; Zeng A *et al.* 2018 Prospectively Isolated

Tetraspanin+ Neoblasts Are Adult Pluripotent Stem Cells Underlying Planaria Regeneration. *Cell* **173**, 1593–1608.e20; Newmark PA, Sánchez Alvarado A. 2000 Bromodeoxyuridine specifically labels the regenerative stem cells of planarians. *Dev. Biol.* **220**, 142–153; Wenemoser D, Reddien PW. 2010 Planarian regeneration involves distinct stem cell responses to wounds and tissue absence. *Dev. Biol.* **344**, 979–991; Cebrià F. 2008 Organization of the nervous system in the model planarian *Schmidtea mediterranea*: an immunocytochemical study. *Neurosci. Res.* **61**, 375–384; Cebrià F, Newmark P a. 2005 Planarian homologs of netrin and netrin receptor are required for proper regeneration of the central nervous system and the maintenance of nervous system architecture. *Development* **132**, 3691–3703; King RS, Newmark PA. 2013 In situ hybridization protocol for enhanced detection of gene expression in the planarian *Schmidtea mediterranea*. *BMC Dev. Biol.* **13**, 8; Asano Y, Nakamura S, Ishida S, Azuma K, Shinozawa T. 1998 Rhodopsin-like proteins in planarian eye and auricle: detection and functional analysis. *J. Exp. Biol.* **201**, 1263–1271; LoCascio SA, Lapan SW, Reddien PW. 2017 Eye Absence Does Not Regulate Planarian Stem Cells during Eye Regeneration. *Dev. Cell* **40**, 381–391; Sarkar A *et al.* 2019 Serotonin is essential for eye regeneration in planaria *Schmidtea mediterranea*. *FEBS Lett.* **593**, 3198–3209).

Line 104: Please rephrase this sentence to make it clearer: “Neural cells were only present internal to the peripheral longitudinal muscle, but were found on either side of the body wall musculature.”

We have edited this statement to make it clearer.

Is the Statocyst regenerated?

Yes, indeed! This result has been published (Srivastava M, Mazza-Curll KL, Van Wolfswinkel JC, Reddien PW. 2014 Whole-body acoel regeneration is controlled by Wnt and Bmp-Admp signaling. *Curr. Biol.* **24**, 1107–1113).

Line 204: “Although the anterior structure meets the generally accepted definition of the term “brain”, which has been applied to anterior condensations in other acoels, we refrain from using this term to avoid an implication of homology with the brains of other xenacoelomorphs or other bilaterians. In this manuscript, we refer to this structure with its many organized regions as the anterior condensation.” I disagree in this point with the authors and would like to encourage them to revise their decision. The use of a term does not and should not imply a homology proposition per se. Many terms such as larva, nerve cords etc. are descriptive terms, which, when properly described, are useful. There is no way around re-using the term for convergent structures. The bird wing is still a wing as it is the *Drosophila* wing. Otherwise we would have to find a new term for the one or the other. I suggest to maybe reference the definition of a brain here – or shortly describe what the authors mean.

There’s been much debate about whether acoels have a true brain (see Gavilán B, Perea-Atienza E, Martínez P. 2016 Xenacoelomorpha: a case of independent nervous system centralization? *Philos. Trans. R. Soc. Lond. B Biol. Sci.* **371**). We are cautious to apply this term, as some researchers believe that for a structure to be called a brain, it must be ganglionic and contain a cortex of cell bodies that surround a central neuropil. Our methods do not detect this

type of structure, which certainly does not mean that *Hofstenia* does not have it. In *Symsagittifera roscoffensis*, for example, EM studies were used to build a case for a true brain-like organization (see Bery A, Cardona A, Martinez P, Hartenstein V. 2010 Structure of the central nervous system of a juvenile acoel, *Symsagittifera roscoffensis*. *Dev. Genes Evol.* **220**, 61–76).

Other authors apply a different perspective, defining a brain as “the organ made of a conglomeration of nerve cells, highly interconnected, typically associated with sensory receptors, in the anterior (relative to direction of movement) part of the body” (Martinez P, Sprecher SG. 2020 Of Circuits and Brains: The Origin and Diversification of Neural Architectures. *Frontiers in Ecology and Evolution* **8**, 82). Therefore, we do agree with the reviewer that rejecting the term brain entirely is potentially too conservative a perspective. Therefore, we have re-worded to suggest that the anterior condensation is brain-like, to leave open the possibility that *Hofstenia* does have a canonical brain. However, we still utilize the “anterior condensation” term throughout the paper.

Can the confocal stacks be made accessible in a public depository after publication? It would be helpful to also gain a better impression of the muscle/nerual layers in the body wall. Last could in addition be added as supplement movie to the paper.

We have added supplementary movies 1 and 2 to show this. Thank you for this suggestion.

Line 308: “However, whether all diverse neural cell types and structures identified in this study are regenerated remains to be determined, as does the timing of their regeneration.”

We have edited this statement.

Line 338: “neural components encoded in the genome”. Well, there are a multitude of neural components. Please restrict this sentence to the neurotransmitter pathway here.

We changed “neural components” to “neurotransmitter synthesis pathway genes”.

Line 340: “our work suggests that the last common ancestral acoel had a more complex nervous system, molecularly and structurally, than has previously been hypothesized.” I can not follow this statement because I think it is not justified from the data provided. An anterior brain was always proposed as ancestral state for Acoela and the nerve plexus was described before and dates back to the times of Steinböck and earlier. Please support this statement with references if I am wrong.

We agree, an anterior brain has long been assumed to be a characteristic of the ancestral acoel. We have removed this statement, which actually enables the subsequent sentences and paragraphs to highlight that: 1) our work reveals a full complement of neurotransmitter synthesis pathway genes, which had not been shown for xenacoelomorphs before, 2) our co-expression studies reveal multiple neural types including those with co-expression of and 3) our work pushes back the timing of when certain components of the brain evolved (e.g., ventral lobes and ring like commissure). These structures had previously only been reported for the *Crucimusculata* (including *Isodiametra pulchra* and *Symsagittifera roscoffensis*) (Achatz JG,

Martinez P. 2012 The nervous system of *Isodiametra pulchra* (Acoela) with a discussion on the neuroanatomy of the Xenacoelomorpha and its evolutionary implications. *Front. Zool.* **9**, 27).

- Figure 2: Is it possible to use the same color for the same marker. Tyr-tub is green in 2g,h and magenta in i-k.

Updated

- Figure 2: What is the posterior signal in 2b?

The anti-FMRF staining reveals some cells in the posterior of the animal, which are putatively a neural or secretory cell population. We have added a statement in the legend (Figure 2b) to mention this.

- Figure 2: Can the muscle layer be indicated in 2x?

Since we don't introduce muscle staining till the next figure, we think it best to leave this schematic as is. It orients the reader to the organization of the anterior condensation, through which they can understand the relationship of the nervous system to muscle in the following section.

- Figure 3: Can the authors add a lateral view of the confocal stack to illustrate the layers?

We have added this as Supplementary Figure 2.

- Figure 3: What is outside the peripheral muscle layer? How is the epidermis structured?

The epidermis, which is the outermost layer of the animal, lies outside the peripheral muscle layer. We are currently investigating the structure of the epidermis using transgenic animals fluorescently-labeled epidermal and muscle cells. Stay tuned for that manuscript!

Schemes in Figure 2, 3, 5 and Supp. Figure 2 : These figures are a bit too schematic. It would be helpful to outline the cells in these crosssections, maybe with help of published histological sections, the confocal stacks etc. It is hard to understand how many cells form one layer, how the epidermis is composed and what the filling tissue (parenchyme?) is composed of. I suggest to elaborate this scheme a bit more.

We agree with the reviewer that these drawings are highly schematized. We aimed to draw the reader's attention to the major domains of neural organization. In the long term we would like to refine these schematics, but given that in situ hybridization approaches do not reveal cellular morphology and relative placement at high resolution, we have opted to take a conservative approach. We hope to utilize transgenic lines with labeled neurons to ultimately re-draw these accurately with the details the reviewer seeks.